# A meta-learning approach to (re)discover plasticity rules that carve a desired function into a neural network

**Basile Confavreux**
IST Austria
and Centre for Neural Circuits and Behaviour
University of Oxford, UK
basile.confavreux@ist.ac.at

**Friedemann Zenke**
Friedrich Miescher Institute
Basel, Switzerland
and Centre for Neural Circuits and Behaviour
University of Oxford, UK

**Everton J. Agnes**
Centre for Neural Circuits and Behaviour
University of Oxford, UK

**Timothy Lillicrap**
Deepmind, London, UK

**Tim P. Vogels**
IST Austria
and Centre for Neural Circuits and Behaviour
University of Oxford, UK

## Abstract

The search for biologically faithful synaptic plasticity rules has resulted in a large body of models. They are usually inspired by – and fitted to – experimental data, but they rarely produce neural dynamics that serve complex functions. These failures suggest that current plasticity models are still under-constrained by existing data. Here, we present an alternative approach that uses meta-learning to discover plausible synaptic plasticity rules. Instead of experimental data, the rules are constrained by the functions they implement and the structure they are meant to produce. Briefly, we parameterize synaptic plasticity rules by a Volterra expansion and then use supervised learning methods (gradient descent or evolutionary strategies) to minimize a problem-dependent loss function that quantifies how effectively a candidate plasticity rule transforms an initially random network into one with the desired function. We first validate our approach by re-discovering previously described plasticity rules, starting at the single-neuron level and "Oja's rule", a simple Hebbian plasticity rule that captures the direction of most variability of inputs to a neuron (i.e., the first principal component). We expand the problem to the network level and ask the framework to find Oja's rule together with an anti-Hebbian rule such that an initially random two-layer firing-rate network will recover several principal components of the input space after learning. Next, we move to networks of integrate-and-fire neurons with plastic inhibitory afferents. We train for rules that achieve a target firing rate by countering tuned excitation. Our algorithm discovers a specific subset of the manifold of rules that can solve this task. Our work is a proof of principle of an automated and unbiased approach to unveil synaptic plasticity rules that obey biological constraints and can solve complex functions.

# 1   Introduction

Synaptic plasticity is widely agreed to be essential for high level functions such as learning and memory. Its mechanisms are usually modelled with plasticity rules, i.e., functions that describe the evolution of the strength of a synapse. Current experimental techniques do not allow the tracking of relevant synaptic quantities over time at the population level, especially over the duration of learning. Therefore, most plasticity rules in the literature were derived from a few experiments in single synapses *ex vivo*, e.g., spike timing-dependent-plasticity [1–5]. Such rules do not usually construct a specific function or architecture to a network model on their own [6], unless they are carefully combined and orchestrated [7–9]. The link between the function of a network and the low level mechanisms that lead to its structure thus remains elusive.

Here we aim to bridge this gap by deducing plasticity rules from indirect but accessible quantities in the brain: the function of a network (e.g., elicited behaviour, population activity, etc.) or its architecture. Major technical breakthroughs in the field of behavioural neuroscience and connectomics have vastly increased the amount of data for different aspects (or levels) of neuroscience [10–13], and we wondered if we could use these newly available results to deduce how a nervous system is constructed from scratch. Here, we present a meta-learning framework aiming to infer plasticity rules based on their ability to ascribe a desired function or architecture to an initially random neural network model. We present three example cases of rate and spiking neural network models for which such a numeric deduction of plasticity rules can be successfully performed. We point out their current limitations and discuss possible ways forward.

# 2   Related work

The idea to use supervised learning to learn unsupervised (local) learning rules dates back to the early 90s [14, 15] and resurfaced recently with the development of robust numerical optimization methods [16, 17], growing computational resources, and the advent of meta-learning [18, 19], which provides a convenient framework to tackle such questions. In some approaches the focus was to learn unsupervised or semi-supervised rules for representation learning or improved generalisation capabilities [20–24]. Others aim to learn optimizers that can then be used for supervised learning [25]. More Neuroscience-oriented approaches attempt to find which learning rules could implement a biologically plausible version of backpropagation [26, 27]. In contrast to most works described previously relying on numerical optimization to find learning rules, others analytically develop and infer learning rules that can elicit certain biologically inspired functions [7, 8, 28–30].

Overall, our work is complementary to the above work. Specifically, we provide a scalable framework to automatically discover biologically plausible rules that carve specific computational functions into otherwise analytically intractable spiking networks while at the same time relying on interpretable parametrizations suitable for experimental verification.

# 3   Results

As a proof of principle that biologically plausible rules can be deduced by numerical optimization, we show that a meta learning framework was able to rediscover known plasticity rules in rate-based and spiking neuron models.

## 3.1   Rediscovering Oja's rule at the single neuron level

As a first challenge, we aimed to rediscover Oja's rule, a Hebbian learning rule known to cause the weights of a two-layer linear network to converge to the first principal vector of any input data set with zero mean [31]. Towards this end, we built a single rate neuron following dynamics such that

$$y_i = \sum_{j=1}^{N} x_j w_{ij}, \tag{1}$$

where $y_i$ is the activity of the postsynaptic neuron $i$ ($i = 1$ in the single neuron case), $x_j$ is the activity of the presynaptic neuron $j$, and $w_{ij}$ is the weight of the connection between neurons $j$ and $i$. Oja's

rule can be written as

$$\Delta w_{ij} = \eta \left( y_i x_j - y_i^2 w_{ij} \right) \qquad (2)$$

where $\eta$ is a learning rate. To rediscover Oja's rule from a nondescript, general starting point, the search space of possible plasticity rules was constrained to polynomials of up to second order over the parameters of the presynaptic activity, postsynaptic activity and connection strength. This resulted in widely flexible learning rules with 27 parameters $A_{\alpha\beta\delta}$, where each index indicates the power of either presynaptic activity, $x_i^\alpha$, postsynaptic activity, $y_i^\beta$, or weight, $w_{ij}^\delta$,

$$\Delta w_{ij}(\mathbf{A}) = \eta \sum_{\alpha,\beta,\delta=0}^{2} A_{\alpha\beta\delta} x_j^\alpha y_i^\beta w_{ij}^\delta. \qquad (3)$$

In this formulation, Oja' rule can be expressed as

$$A_{\alpha\beta\delta}^{\text{Oja}} = \left\{ \begin{array}{rl} 1, & \text{if } \alpha=1, \beta=1, \delta=0 \\ -1, & \text{if } \alpha=0, \beta=2, \delta=1 \\ 0, & \text{otherwise.} \end{array} \right. \qquad (4)$$

Note that higher order terms could be included but are not strictly necessary for a first proof of principle. We went on to investigate if we could rediscover this rule from a random initialisation. We simulated a single rate-neuron with $N$ input neurons and an initially random candidate rule in which parameters were drawn from a Gaussian distribution, $A_{\alpha\beta\delta} \sim \mathcal{N}(0, 0.1)$ (Fig. 1A). A loss function was designed by comparing the final connectivity of a network trained with the candidate rule to the first principal vector of the data, $\mathbf{PC_1}$, i.e., we aimed for a rule able to produce the first principal vector, but were agnostic with respect to the exact parametrisation,

$$\mathcal{L}(\mathbf{A}) = \langle \| \mathbf{w} - \mathbf{PC_1} \| \rangle_{datasets}. \qquad (5)$$

The norm used throughout this study is the $L^2$ norm. Updates of the plasticity rule were implemented by minimizing $\mathcal{L}$ using the Covariance Matrix Adaptation Evolution Strategy (CMA-ES) method [17]. We chose CMA-ES instead of a gradient based strategy such as ADAM [16]due to its better scalibilty with network size (Supplementary Fig. S1). Clipping strategies to deal with unstable plasticity rules (rules that trigger a numerical error in the inner loop and thus an undefined loss) were used. To prevent overfitting (rules that would score low losses only on specific datasets), each plasticity rule was tested on many input datasets from a $N$-dimensional space (typically $N_{dataset} = 20$). Overall, this approach successfully recovers Oja's rule, for any input dataset dimensions that we could test in reasonable time (up to 100 input neurons, Fig. 1B and Fig. 1C).

## 3.2 Rediscovering two co-active plasticity rules in a rate network

Next, we extended our framework to the network level, using a two-layer rate network, in which $M \leq N$ interconnected output neurons extracted further principal components from a $N$-dimensional input dataset [32] (Fig. 2A). To extract additional principal components, this network should use Oja's rule to modify the input (feedforward synapses), and the lateral connections between neurons of the output layer should be adjusted by an anti-Hebbian learning rule [32]. In our model, in addition to the all-to-all connections between input and output layers, the output neurons were thus interconnected with plastic lateral connections, being described by

$$y_i = \sum_{j=1}^{N} x_j w_{ij} + \sum_{j=1}^{M} y_j \hat{w}_{ij}, \qquad (6)$$

where $\hat{w}_{ij}$ is the lateral connection between output neurons $j$ and $i$. The desired network function (that determines our loss function) expressed the first $M$ principal vectors of the input dataset in the activity of the output neurons.

In our framework, both the feedforward and the lateral plasticity rules were parameterised like in the single neuron case above. Specifically, the additional *lateral* plasticity rule acting on synapses within the output layers followed

$$\Delta \hat{w}_{ij}(\mathbf{B}) = \eta_l \sum_{\alpha,\beta,\delta=0}^{2} B_{\alpha\beta\delta} y_i^\alpha y_j^\beta \hat{w}_{ij}^\delta. \qquad (7)$$

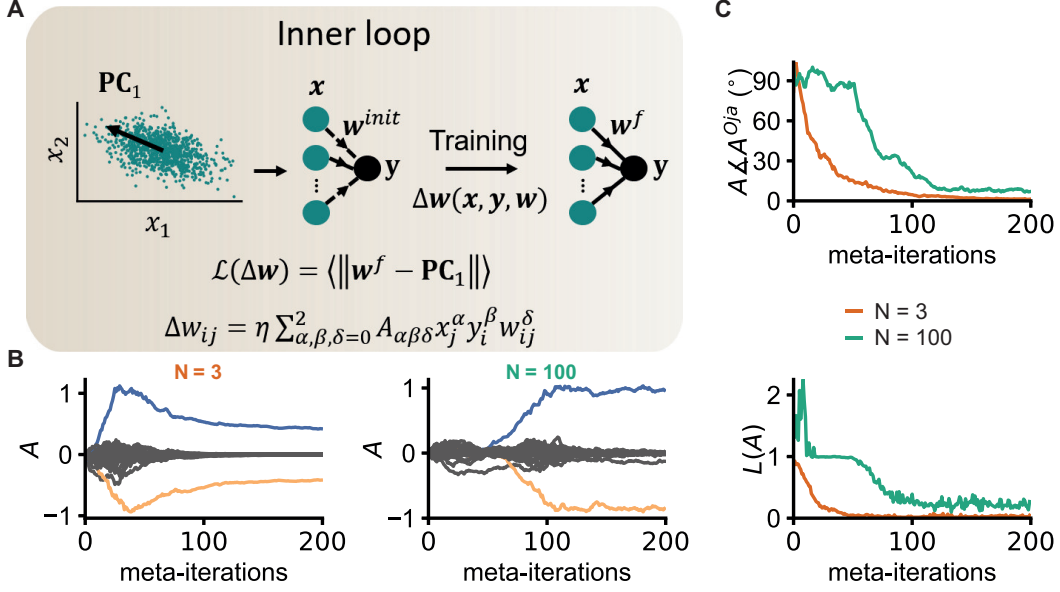

Figure 1: **Rediscovering Oja's rule in a single rate-neuron. A,** In the inner loop, test-networks associate a scalar -the loss-, to any plasticity rule, reflecting how well the plasticity rule considered ascribes a desired function to a two-layer linear network. The network function considered here is having the weights of the network converge to the first principal vector of a given training dataset. The training sets are drawn from multivariate Gaussian distributions centered at the origin. All the training datasets used throughout an optimization process have a covariance matrix originating from a single diagonal matrix which is itself then randomly rotated. The test-networks were trained using batch learning ($N_b = 200$), and an additional learning rate for the weight updates ($\eta = 1/20$). The set of possible plasticity rules considered approximates any dependency of weight updates in the current synaptic strength, pre and post synaptic activity (no time dependency). In the outer loop, the loss computed by the test-networks is optimized using CMA-ES to find (locally) optimal rules for the desired function. **B,** Evolution of every coefficient $A_{\alpha\beta\delta}$ of the candidate plasticity rule during two optimization processes: one with 3 inputs neurons, the other with 100 (a single post-synaptic neuron in both cases). The parameter plotted in blue corresponds to $A_{110}$ ($A_{110}^{\text{Oja}} = 1$), the one in yellow to $A_{021}$ ($A_{021}^{\text{Oja}} = -1$). **C,** Evolution of the loss and of the angle between the vector representing Oja's rule ($\mathbf{A}^{\text{Oja}}$) and the candidate plasticity rule ($A$) during both optimization processes plotted in B.

In this formulation, the correct anti-Hebbian rule could be parameterised as

$$B_{\alpha\beta\delta}^{\text{antiH}} = \begin{cases} -1, & \text{if } \alpha = 1, \beta = 1, \delta = 0 \\ 0, & \text{otherwise.} \end{cases} \tag{8}$$

It should be noted here that the structure of the lateral connectivity within the output layer was fixed and hierarchical (Fig. 2A), such that some neurons only sent, or only received synapses. Only the weights of the existing connections were changed according to the candidate plasticity rules described above. Both candidate plasticity rules were co-optimized using the same optimizer as previously described [17]. A loss function was designed to quantify how much the incoming weights to each output neuron differed from the $i^{th}$ principal component,

$$\mathcal{L}(\mathbf{A}, \mathbf{B}) = \sum_{i=1}^{M} \langle \| [w_{i1} \quad w_{i2} \quad \ldots \quad w_{iN}] - \mathbf{PC}_i \| \rangle_{datasets}, \tag{9}$$

where $[w_{i1} \quad w_{i2} \quad \ldots \quad w_{iN}]$ are the incoming connections to the $i^{th}$ output neuron and $\mathbf{PC}_i$ the $i^{th}$ principal vector of the input dataset after training.

Our algorithms were able to recover both target plasticity rules – Oja's rule and the anti Hebbian rule – up to a scalar factor (which could be attributed to the $L^1$ regularization used on the coefficients

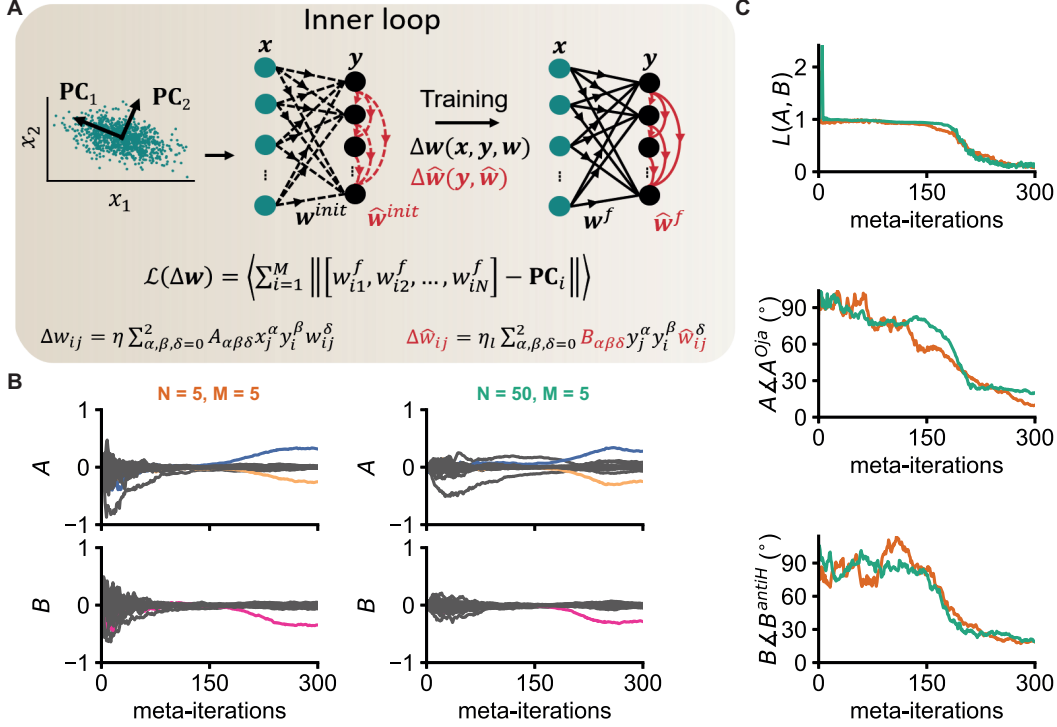

Figure 2: **Rediscovering Oja's rule and an anti-Hebbian rule. A,** In a similar fashion to Fig. 1, a two-layer linear rate network is used in the inner loop, this time with additional output neurons. The between layer connections are all-to-all and evolve according to $\Delta \mathbf{w}$. The connections within the output layers are hierarchical by design (not learnt) and follow $\Delta \hat{\mathbf{w}}$. Both plasticity rules are drawn from a polynomial expansion on the current weights, pre- and postsynaptic activity, similar to Fig. 1. The loss computed in the inner loop quantifies how well the connections to each output neuron match the $i^{th}$ principal component of the input dataset after training (ordering based on the hierarchical connections). The datasets are generated similarly to Fig. 1. The test-networks were trained using batch learning ($N_b = 200$), and an additional learning rate for each plasticity rule ($\eta = 1/20$ and $\eta_l = 1/10$) hand-tuned on a network using the optimal plasticity rules. The inner loop parameters were found by minimizing compute time to obtain a small loss with the target plasticity rules. CMA-ES was then used on the loss with respect to the parameters of the two plasticity rules. **B,** Evolution of every coefficient $A_{\alpha\beta\delta}$ and $B_{\alpha\beta\delta}$ of the candidate plasticity rule during two optimization processes: one with 5 inputs neurons and 5 output neurons, the other with 50 input neurons and 5 output neurons. The parameter plotted in blue corresponds to $A_{110}$ ($A_{110}^{\text{Oja}} = 1$), the one in yellow to $A_{021}$ ($A_{021}^{\text{Oja}} = -1$) and the one in pink to $B_{110}$ ($B_{110}^{\text{antiH}} = -1$). **C,** Evolution of the loss and of the angle between the vector representing Oja's rule -$A^{Oja}$- (respectively the target anti Hebbian rule -$B^{antiH}$-) and the candidate plasticity rule -$A$- (respectively $B$) during both optimization processes plotted in panel B.

of the plasticity rule) in networks of up to 50 input and 5 output neurons (Fig. 2B and Fig. 2C). Increasing the size of the networks in the inner loop exponentially increases the computing time until convergence of the meta-optimization. We observed that, for larger network sizes, more meta-iterations were needed, while each iteration in the inner loop required longer to compute, and more iterations were required for the networks in the inner loop to converge.

### 3.3 Learning inhibitory plasticity in a spiking neuron

In a first attempt to produce candidate rules that could inspire and guide future experimental studies of natural, multi-cell-type plasticity, we introduced a more biologically realistic neuron model, i.e., a model that produced spikes. Following previous work [29], we constructed a model with a single conductance-based leaky integrate-and-fire neuron, receiving 800 excitatory and 200 inhibitory

afferents that are separated in 8 input groups (Fig. 3A). Excitatory afferents were hand-tuned, with a set of strengthened, preferred signal afferents. Inhibitory afferents were plastic and initially random.

The membrane potential dynamics of the postsynaptic neuron followed

$$\tau_\mathrm{m}\frac{\mathrm{d}V(t)}{\mathrm{d}t} = -(V(t) - V_\mathrm{rest}) - \frac{g_\mathrm{E}(t)}{g_\mathrm{leak}}(V(t) - E_\mathrm{E}) - \frac{g_\mathrm{I}(t)}{g_\mathrm{leak}}(V(t) - E_\mathrm{I}),\tag{10}$$

with the excitatory and inhibitory conductances $g_\mathrm{E}(t)$ and $g_\mathrm{I}(t)$ for excitatory and inhibitory synapses, respectively. A postsynaptic spike is emitted whenever the membrane potential $V(t)$ crosses a threshold $V_\mathrm{th}$ from below, with an instantaneous reset to $V_\mathrm{reset}$. The membrane potential is clamped at $V_\mathrm{reset}$ for the duration of the refractory period, $\tau_\mathrm{ref}$, after the spike. Conductances changed according to

$$\frac{\mathrm{d}g_\mathrm{E}(t)}{\mathrm{d}t} = -\frac{g_\mathrm{E}(t)}{\tau_\mathrm{E}} + \bar{g}_\mathrm{E}\sum_k w_k^\mathrm{E}S_k(t) \quad \text{and} \quad \frac{\mathrm{d}g_\mathrm{I}(t)}{\mathrm{d}t} = -\frac{g_\mathrm{I}(t)}{\tau_\mathrm{I}} + \bar{g}_\mathrm{I}\sum_k w_k^\mathrm{I}(t)S_k(t),\tag{11}$$

where $\tau_\mathrm{m} = 20$ ms, $V_\mathrm{rest} = -60$ mV, $E_\mathrm{E} = 0$ mV, $E_\mathrm{I} = -80$ mV, $g_\mathrm{leak} = 10$ nS, $\bar{g}_E = 0.014 g_\mathrm{leak}$, $\bar{g}_I = 0.035 g_\mathrm{leak}$, $\tau_\mathrm{E} = 5$ ms, and $\tau_\mathrm{I} = 10$ ms were taken from previous work [29], and $S_k(t) = \sum \delta(t - t_k^*)$ is the spike train of presynaptic neuron $k$, with $t_k^*$ being the spike times of neuron $k$. The variables $x_k(t)$ and $x_{post}(t)$ account for the trace of spike trains of pre- and postsynaptic spikes,

$$\frac{\mathrm{d}x_k(t)}{\mathrm{d}t} = -\frac{x_k(t)}{\tau_\mathrm{pre}} + S_k(t) \quad \text{and} \quad \frac{\mathrm{d}x_\mathrm{post}(t)}{\mathrm{d}t} = -\frac{x_\mathrm{post}(t)}{\tau_\mathrm{post}} + S_\mathrm{post}(t),\tag{12}$$

where $\tau_\mathrm{pre}$ and $\tau_\mathrm{post}$ are the time constants of the traces associated to the pre- and postsynaptic neurons, respectively.

The input spike trains were generated similarly to previous work [29]. We defined 8 input groups, each having a time varying firing rate added to a baseline of 5 Hz. The varying firing rate consisted of a random walk with $\tau_\mathrm{input} = 50$ ms, followed by a sparsification of the number of activity bumps above baseline, in which every second bump of activity was omitted (see ref. 29 for details).

Only the inhibitory synaptic weights, i.e., $w_k^\mathrm{I}(t)$ were plastic. Excitatory weights were defined according to their input group,

$$w_k^\mathrm{E} = 0.3 + \frac{1.1}{(1 + (G_k - P))^4} + \epsilon_k,\tag{13}$$

where $G_k$ is the input group of afferent $k$, $P = 5$ is the preferred input group, and $\epsilon_k$ is a noise term drawn from uniform distribution between 0 and 0.1. The candidate plasticity rule was parametrised using pre- and postsynaptic spike times,

$$\frac{\mathrm{d}w_k^\mathrm{I}(t)}{\mathrm{d}t} = \alpha S_k(t) + \beta S_\mathrm{post}(t) + \gamma x_k(t)S_\mathrm{post}(t) + \kappa x_\mathrm{post}(t)S_k(t).\tag{14}$$

The average changes of the rule can be described as a Volterra expansion of synaptic changes based on the activity of pre- and postsynaptic activities, $\nu_\mathrm{pre}$ and $\nu_\mathrm{post}$, respectively,

$$\left\langle\frac{\mathrm{d}w^\mathrm{I}}{\mathrm{d}t}\right\rangle = \alpha\nu_\mathrm{pre} + \beta\nu_\mathrm{post} + \gamma\tau_\mathrm{pre}\nu_\mathrm{post}\nu_\mathrm{pre} + \kappa\tau_\mathrm{post}\nu_\mathrm{post}\nu_\mathrm{pre}.\tag{15}$$

Following previous work [29], we aimed for a plasticity rule to establish stable postsynaptic firing rate $r_\mathrm{tg} = 5$ Hz (Fig. 3A). As such we expressed the loss function as

$$\mathcal{L}(\tau_\mathrm{pre}, \tau_\mathrm{post}, \alpha, \beta, \gamma, \kappa) = \left\langle\frac{(\bar{r} - r_\mathrm{tg})^2}{\bar{r} + 0.1}\right\rangle_\mathrm{datasets},\tag{16}$$

where $\bar{r}$ is the postsynaptic firing rate measured during a 10 seconds window, chosen randomly inside a 30 seconds scoring phase after a 1 minute training phase. The denominator penalized very low firing rates and thus aided the optimization.

For the learning rule to enforce a stable (and low) firing-rate, we expected the framework to produce a learning rule that balances excitation and inhibition, similar to previously proposed, hand-tuned spike-timing based inhibitory plasticity rules [29, 33], in which EI balance was shown to be a by-product

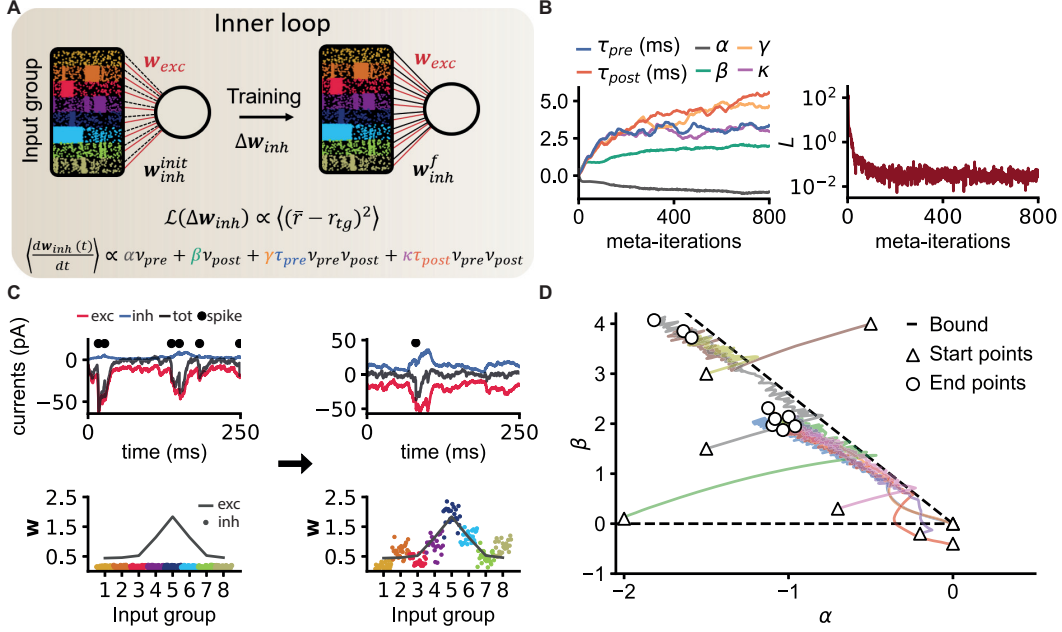

Figure 3: **Inhibitory plasticity in a spiking neuron. A,** A single conductance-based leaky integrate and fire neuron receives tuned excitatory and inhibitory inputs. 8 inhomogeneous Poisson processes were generated (as in [29]), with 100 excitatory and 25 inhibitory spike trains drawn from each process. The excitatory weights were fixed to a shared group value. The inhibitory weights were initialised randomly. After 1min, the number of spikes in a 10s window chosen randomly within a 30s period was used to devise a loss quantifying the difference between the neuron's firing rate and the desired target firing rate. A memory of previous spikes is kept through a synaptic trace $x$ associated to every neuron. To prevent overfitting, each plasticity rule is tested $N_{datasets}$ different input spike trains. **B,** The evolution of each parameter of the candidate plasticity rule, along with the associated loss is plotted across meta-iterations. **C,** To illustrate the effect of the rule learnt in **B**, postsynaptic currents and inhibitory weights are plotted both at the start of a simulation and after one minute during which the inhibitory weights evolved according to the learnt rule. The grey line corresponds to the mean group values of the excitatory weights. These weights are rectified to account for the difference in driving force compared to the inhibitory weights. The inhibitory weights were set to be initially weak. **D,** Optimization trajectories are plotted in the $(\alpha, \beta)$ subspace of the plasticity rule parameters. The dotted line show two extreme steady state solutions with $\mathcal{L} = 0$: the horizontal line corresponds to Hebbian terms only ($\beta = 0$), while the other line uses only the postsynaptic term ($\gamma \tau_{\text{pre}} + \kappa \tau_{\text{post}} = 0$).

of a learning rule imposing stable (constant) firing-rates. The family of rules found using ADAM [16] with gradients computed using finite differences on the search space defined in Eq. 14 differs from the rules reported previously [29] (Fig. 3B). The previously published inhibitory plasticity rule [29] relies solely on a presynaptic decay term and a Hebbian term to establish target firing (and EI balance). The rules found here use a mixture of Hebbian terms ($\gamma, \tau_{\text{pre}}, \kappa, \tau_{\text{post}}$) and sole postsynaptic terms ($\beta$). This can be explained by a steady state analysis of the model, showing that

$$\nu_{\text{post}} = -\frac{\alpha \nu_{\text{pre}}}{\beta + (\gamma \tau_{\text{pre}} + \kappa \tau_{\text{post}}) \nu_{\text{pre}}}. \tag{17}$$

This means that the six parameters of the plasticity rule are bound by a single equation, and can compensate for each other, so there are many more ways to achieve a target firing rate than relying on Hebbian terms only [29, 33], or on postsynaptic terms only. The parameters $\alpha$ and $\beta$ are constrained to be negative and positive, respectively, to ensure that the fixed-point for the firing-rate is stable.

Our algorithm always started the meta-learning process by balancing $\alpha$ and $\beta$, because that is effectively the quickest way to decrease the loss. Only in a second step the algorithm optimised other parameters that give additional, albeit smaller benefits. Mathematically, we can understand this behaviour from the fact that the combination of the Hebbian learning rates, $\gamma$ and $\kappa$, multiplied by

the time constants of the traces, $\tau_{\text{pre}}$ and $\tau_{\text{post}}$, automatically sets $\beta$ to dominate the denominator of the equation above ($\beta >> \gamma\tau_{\text{pre}} + \kappa\tau_{\text{post}}$). This creates a bound in the possible learning rules,

$$\beta = -\frac{\alpha\nu_{\text{pre}}}{\nu_{\text{post}}}. \tag{18}$$

Our intuition is confirmed when plotting several optimization trajectories in the $(\alpha, \beta)$ plane (Fig. 3D). We noticed that the family of rules found by our framework all stayed closed to the boundary, reflecting the small subset of solutions selected by the algorithm, both relatively "easy" to find in parameter space and quick to establish steady state.

The manifold of compliant plasticity rules is bigger than Eq. 18 implies (i.e., that only $\alpha, \beta$ matter). This can be seen, e.g., in rules that make use of the Hebbian terms at their disposal. However, due to long compute times, we only allowed one minute of simulated time to elapse for a postsynaptic neuron to enforce the desired firing rate, therefore the steady state solutions that are too slow to appear don't have as good a loss with our framework. It follows that non-zero $\beta$ is an efficient way to quickly establish the desired firing rate in our set-up. Notably, non-zero Hebbian terms allow the establishment of a *detailed* balance [34] (Fig. 3C), in a similar fashion to previously studied inhibitory plasticity rule [29] (Fig. 3C, see Supplementary Fig. S2 for more details). They allow a more regular firing and thus help the loss to be more reliable. Such elements to the rules could become more important in recurrent network simulations.

## 4   Discussion

We propose a meta-learning approach that searches and finds (locally) optimal plasticity rules with respect to a desired network function or architecture. Our framework requires both the ability to quantify the desired network function through the design of a loss, and a sensible yet reasonably flexible parametrisation of the candidate plasticity rules, and the quantities and variables they must rely on. Our framework is able to recover known rules in both rate and spiking models.

A recent approach with similar aims predicts testable plasticity rules in spiking neurons, but uses a different optimization strategy (cartesian genetic programming) and subsequently a different parametrisation [35], and is thus complementary to our work.

However, several challenges remain: our approach is computationally heavy, it remains to be studied how it fares in large non linear systems like spiking networks. Even though we are interested in the learning rules themselves (which should be relatively scale invariant) and not in large networks per se, problems that cannot be down scaled efficiently could remain out of reach. Moreover, the parametrisations used in this study, while flexible enough for a proof of principle, might need to be extended to describe real-life plasticity rules.

## 5   Conclusion and future work

In summary, we present a proof of principle that plasticity rules can be derived with a meta-learning framework that iteratively refines new rules through minimisation of a loss function reflecting a desired network output. There are multiple challenges ahead, ranging from technical issues of computing gradients (or not) to the choice of a parametrisation but our results promise a new perspective on plasticity rules that may explain both form and function of cortical circuitry.

## Broader Impact

There may be up to 140 different synaptic plasticity rules at play in everyday behaviours such as making a simple memory. We have only begun to understand five or less of these rules, and for the foreseeable future experimental neuroscience will not be able to deliver the necessary data to dis-entwine this difficult puzzle. Machine learning and modern computing, on the other hand, have made huge advances in being able to simulate and analyse highly complex tasks. Utilising this power to infer plasticity rules and thus create experimental hypothesis is entirely possible, timely and urgent. We thus propose a first step in the development of a set of computational tools that allows us to discover the synaptic plasticity mechanisms responsible for developing and maintaining complex

structures through neuronal activity. Machine learning techniques give us the benefit of targeted, gradient-directed searches combined with fast and computationally powerful searches. We aim to eventually run our meta-learning algorithms to achieve connectivity and function of healthy and aberrant neural phenomena. Soon, we will be able to directly affect translational approaches that aim to utilise plasticity protocols for therapeutic approaches. Finding the families of plasticity rules that create functional neuronal networks in the brain will be a crucial and long lasting contribution to basic and applied science. Finally, our findings may also inspire the development of new ML tools, both for the analysis and training of artificial neural networks, which still have to live up to their potential in terms of generalisation and semantic knowledge representation. Biologically inspired rules may just prove to be the solution to many a problem at hand.

## Acknowledgments and Disclosure of Funding

We would like to thank Chaitanya Chintaluri, Georgia Christodoulou, Bill Podlaski and Merima Šabanović for useful discussions and comments. This work was supported by a Wellcome Trust Senior Research Fellowship (214316/Z/18/Z), a BBSRC grant (BB/N019512/1), an ERC consolidator Grant (SYNAPSEEK), a Leverhulme Trust Project Grant (RPG-2016-446), and funding from École Polytechnique, Paris.

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
