[Supplementary Material]

# Supplementary material: A meta-learning approach to (re)discover plasticity rules that carve a desired function into a neural network

## 1 Comparison of gradient-based optimization and evolutionary strategies

Two optimizers were tested in this study: a gradient-based using ADAM with gradients of $\mathcal{L}$ with respect to the coefficients of the plasticity rule computed with finite differences, and a Covariance Matrix Adaptation Evolutionary Strategy (CMA-ES). In our work, ADAM scaled poorly with the network size, in contrast to CMA-ES, which led us to use the latter for the larger networks (Fig. 1 and Fig. 2). Indeed, ADAM failed to discover Oja's rule for networks with more than $N = 11$ inputs. To investigate why finding Oja's rule was less successful for larger $N$, we plotted the loss landscape along an ideal optimization trajectory in two input dimensions (Fig. S1A). Stable regions (corresponding to rules that can train networks without triggering numerical errors) get "smaller" for higher $N$, i.e., increasingly localized around the subspace generated by Oja's rule (Fig. S1B and Fig. S1C). This finding supports the choice of local numerical methods like ADAM or CMA-ES, as such methods can always be initialized at the origin which is stable for any input dimension. Overall the poor scaling of gradient-based optimization is not just due to the linearly increasing computing time that prevents a more thorough hyperparameter tuning for larger $N$, but the loss landscape itself becomes increasingly less well-behaved in higher dimensions (Fig. S1C). Given the relatively low number of parameters in our set-ups (<100), an evolutionary strategy outperforming a gradient based one isn't a surprising find.

Figure S1: **Loss landscape. A,** Plot in the vector space representing all the plasticity rules considered in this study. The loss along the straight line between a random initial plasticity rule $\mathbf{A}^0 \sim \mathcal{N}(0, 0.1)$ and Oja's rule, $\mathbf{A}^{\text{Oja}}$, is plotted in the next subfigure. In addition, a random plane containing the line $(\mathbf{A}^0, \mathbf{A}^{\text{Oja}})$ is chosen and three equally spaced segments belonging to this plane and orthogonal to $(\mathbf{A}^0, \mathbf{A}^{\text{Oja}})$ are chosen. **B, C,** Loss along the segments described above using 3-dimensional (respectively 39-dimensional) input datasets. Flat regions above the dotted gray line in the loss correspond to unstable rules (numerical error in the inner loop) and the use of gradient clipping.

## 2 Analysis of the spike-based inhibitory rules

To motivate the plotting of optimization trajectories in the $(\alpha,\beta)$ plane (Fig. 3D), we show the effect on membrane currents and weights distribution of three plasticity rules: the iSTDP rule from ref. 1 (Fig. S2A), the rule plotted in Fig. 3B, learnt with our framework (Fig S2B), and finally an ablated version of the previous rule, having removed the Hebbian terms (keeping only $\alpha$ and $\beta$) (Fig S2C). Both the Vogels rule and our learnt rule have a similar effect on the weights and create a detailed balance (Fig S2A,B): the inhibitory weights match the excitatory weights on a per-group basis. The final inhibitory weights with the ablated rule are substantially different: the inhibitory weights from all groups converge to the same value, i.e. the detailed balance is lost. This is not surprising as the detailed balance balance arises from the Hebbian terms. However, this ablated rule scores a similar loss to the two others. This implies that our learnt rule solves the task mainly through $\alpha$ and $\beta$, with the Hebbian terms only providing a minor helping to make the loss more reliable. The exact solutions shown in Fig. 3D may depend on the training protocol, e.g., the length of the scoring window, or the time constant of the input statistics.

Figure S2: **Comparison of plasticity rules for spiking set-up.** The inhibitory weights are plotted individually, the color corresponding to which group the synapse belongs to. The gray line corresponds to the mean group value for the excitatory weights (fixed). The right plots show the weights after two minutes of simulation with either the iSTDP rule from Vogels et al. 2011, the rule plotted in Fig. 3B, or a similar rule with the Hebbian terms set to 0. The loss achieved by all three plasticity rules all have comparable values. The middle plots are a visualization of the three plasticity rules. The orange (blue) line show the weight update during a postsynaptic (presynaptic) spike, depending on how long ago was the last presynaptic (postsynaptic) spike.

## References

[1] Tim P Vogels, Henning Sprekeler, Friedemann Zenke, Claudia Clopath, and Wulfram Gerstner. Inhibitory plasticity balances excitation and inhibition in sensory pathways and memory networks. *Science*, 334:1569–1573, 2011.