[Reviews · NeurIPS 2020]

Review 1

Summary and Contributions: The paper proposes a meta-learning approach to find explainable plasticity rules that install a certain functionality in a neuron or network, where the plasticity rule itself is given as a mathematical expression. The main contributions are the method in itself, and empirical evidence supporting that the proposed method works as intended for small problems with both rate-based neuron models as well as spike-based neuron models.

Strengths: One strength of the proposed approach is that, in principle, it could allow to find synaptic plasticity rules that can install a certain functionality on the network level among the set of plasticity rules that are considered biologically realistic. An additional strength is that these plasticity rules are described by symbolic formulas, and because they are optimised using L1 regularisation, the resulting plasticity rules facilitate explainability due to a smaller number of terms.

Weaknesses: There are some weaknesses within the proposed approach, many of which are already identified and acknowledged by the authors. For example, the approach apparently does not scale well. This seems to be due to the fact that finite-difference gradients are used for optimisation, and due to a loss landscape that is very unforgiving. In order to yield meaningful insights into brain functioning and plasticity on the network level, the network needs to be somewhat larger than a handful of neurons, which I think is somewhat a problematic issue with the present approach.

Correctness: The claims are well-supported by the experiments, which appear to be carried out correct. The authors provide insightful figures that further support correctness of methodology.

Clarity: The paper is well-understandable and clearly written.

Relation to Prior Work: Prior work is adequately addressed but one could try to add a paragraph to connect it to relevant works in terms of meta-learning in machine learning. For instance, in [1] potential weight changes, obtained by BP, are further transformed by a meta-learned rule, or [2] learns how much plasticity should be exhibited by certain synapses. However, I am not aware of a similar approach that considers symbolic weight update rules, hence further supports the novelty of this work. [1] Andrychowicz, M., Denil, M., Gomez, S., Hoffman, M. W., Pfau, D., Schaul, T., ... & De Freitas, N. (2016). Learning to learn by gradient descent by gradient descent. In Advances in neural information processing systems (pp. 3981-3989). [2] Miconi, T., Clune, J., & Stanley, K. O. (2018). Differentiable plasticity: training plastic neural networks with backpropagation. arXiv preprint arXiv:1804.02464.

Reproducibility: Yes

Additional Feedback: I believe that the paper is well-written and does present its results adequately. My main concern is that the approach is stuck at the level of just a few neurons, and it is not clear if it would ever apply to more complicated scenarios, where there is opportunity to really discover new plasticity rules that achieve learning in such networks of neurons. l.100: I think "...received synapses" should be "...receive." l.160: "we only allowed *one* minute" Post rebuttal update: The authors have responded to my concern with regard to the scalability of the approach with a different optimization procedure, showing that it applies to larger problems. Hence, I increased my score.


Review 2

Summary and Contributions: The authors used meta-learning to train synaptic plasticity rules. They were able to recover two known plasticity rules in a simple but classical use case. SGD was able to identify Oja’s rule and an anti-Hebbian rule in neural networks performing PCA. They also applied this method to an excitatory-inhibitory spiking circuit, and found that the inhibitory plasticity rules emerged live in an analytically predicted region in the parameter space.

Strengths: Identifying the set of synaptic plasticity rules that support development, learning, and memory in brains remains one of the most daunting challenge in neuroscience. A success at this effort could presumably benefit machine learning as well. So the overall topic is important, and the authors are testing recent machine learning tools to this classical problem. This work is relevant to the NeurIPS community.

Weaknesses: The main weakness of this work, in my opinion, is its lack of empirical demonstration that this approach could actually scale and generalize. The authors first focused on a very simple problem, recovering Oja’s rule, and already demonstrated that in a modestly-sized network (# input neurons > 11), it is difficult to do so. This of course doesn’t suggest that the entire meta-learning approach to synaptic plasticity rules couldn’t scale, rather it concerns the specific implementation and parametrization used by the authors. In the second example, the author shows that the meta-learning approach recovers plasticity rules (Fig 4D) that could be derived with one-line of derivation (eq 13 to eq 15). So here it’s not clear what the additional value of the meta-learning approach is. ---------- Update after rebuttal: I have updated my score from 5 to 6 after seeing the new results on better scalability.

Correctness: The empirical methods are sound. Typo at line 153, should be $\beta = -\alpha \niu_pre / \niu_post$

Clarity: The paper is overall clearly written and understandable. It is, however, not very clear to me how in section 2.3, the authors were able to train the plasticity rule of a spiking network (a non-differentiable system) using gradient-based methods.

Relation to Prior Work: This work is well-motivated and well-referenced. A potentially relevant reference missing is Miconi, Clune, Stanley 2018.

Reproducibility: Yes

Additional Feedback: None.


Review 3

Summary and Contributions: The authors propose to learn the meta-parameters of unsupervised learning rules by performing supervised learning on top of training a neural network on various datasets.

Strengths: This work offers a proof of principle for the learning of meta-parameters. This approach complements existing analytical work relating learning rules to computational objectives.

Weaknesses: The significance of this work for neuroscience is not clear

Correctness: Yes

Clarity: Yes

Relation to Prior Work: The gap between the learning rules and the computational function has been bridged previously by analytical derivations of online optimization algorithms, see e.g. C. Pehlevan and D. B. Chklovskii, "Neuroscience-Inspired Online Unsupervised Learning Algorithms: Artificial Neural Networks," in IEEE Signal Processing Magazine, vol. 36, no. 6, pp. 88-96, Nov. 2019, doi: 10.1109/MSP.2019.2933846.

Reproducibility: Yes

Additional Feedback: Given that analytical derivations of Oja-like learning rules (considered in the first two problems) exist, the implication of their work for neuroscience is not clear. Such analytical derivations from principled objective functions that reflect computational tasks are not subject to the limitations of numerical optimization reported here such as having only a few input or output channels (in contrast to thousands in a biological neural network). Is their optimization algorithm modeling the evolution of learning rules? If yes, they should discuss this. If not, how can it be used to gain insight into brain computation? What specific predictions can be tested experimentally? When, in general, do they expect numerical optimization to be needed? I find the third problem addressed in the paper somewhat artificial because it assumes that the excitatory synapses are stable while inhibitory - plastic. It is my impression that in biology the opposite relationship exists. If so, their model may miss the dynamics of synaptic weights taking place in biological neurons.

[Author Response · NeurIPS 2020]

We thank the reviewers for their thorough reading and thoughtful comments. Below we address their major concerns. All minor points are corrected in the revised draft.

Reviewers R1 & R2 agreed with some of the concerns we raised in the original submission regarding **scalability and generalization** of our results (old Fig. 2). Since submission, we have made substantial progress in scaling our approach and networks. We now use an evolutionary search strategy (i.e., CMA-ES; Hansen, arXiv 2016) in the outer loop of our algorithm, to replace SGD and finite differences. CMA-ES has significantly improved scaling properties and allowed us reproduce our results with larger (up to N=100) rate networks (see Figure below) and substantially shortened computing times for both rate and spiking neuron simulations (not shown here). We will include these new methods and results in our manuscript. The rapidly expanding optimization literature will allow us to meta-learn plasticity rules for problems of larger scale and complexity (e.g., via reinforcement learning strategies, such as Abdolmaleki et al. arXiv 2018). This strategy has already greatly improved the performance of our code. The next challenge will be developing a tool to go beyond single GPU computers to simulate many of such networks in parallel. Even if questions such as how to make a spiking network learn and recall stimuli are difficult problems to date, we are confident that they must be tackled. Our work presents a first avenue towards this end. It should be noted that scalability is idiosyncratic, i.e., specific to the problem rather than framework-wide, and challenges will change for network-level problems. For example, the loss function to discover Oja's rule aims to match every synaptic weight to a desired value. This is not required for other problems, and the loss-landscape for a network task may thus be easier, or at least *different* but not necessarily more difficult to navigate.

Reviewers R2 & R4 also raised questions about the **relevance of our work for neuroscience and additional value of a meta-learning approach.** Uncovering the plasticity rules that underlie brain functions such as learning or memory is a key and open question in neuroscience. Plasticity rules directly deduced from experimental data have thus far failed to exhibit rules that accomplish goals useful for the brain on their own. Meta-learning approaches similar to those develop here could help bridge the gap between empirical observations and function. Our study is a proof of principle to show that known/analytically predictable rules can be successfully recovered by a meta-learning approach on a biologically interpretable search space of plasticity rules.

Analytical approaches deducing a small number of plasticity rules that optimally implement a given function have also shown to be successful (Pehlevan and Chklovskii IEEE 2019). However, one cannot be sure that the learning rules used by the brain are analytically tractable, or which functions are being optimized by the brain -if any-. The dizzying number of different synapse types and thus learning rules at play in the brain make alternative, numeric strategies imperative, because we can also deduce rules that don't necessarily behave *well*. The ultimate goal of our approach is to find learning rules from large datasets, by comparing, e.g., patterns of activation before and after learning, and asking by what rules a simulated network has to be altered to achieve comparable dynamics. Such goals seem within reach. From the ML perspective, the search for unsupervised learning rules able to produce useful representations that could provide alternatives to backpropagation is an active area of research (Jaderberg et al. ICML 2017). Our approach provides real-world challenges that sit at the interface of neuroscience and could contribute to such alternatives.

**In summary**, our approach is complementary to both analytical and experimental ones, as we can apply any loss function *in silico*, e.g., functions that incorporate experimental data, desired behavioral outputs or functional hypotheses. Our meta-learning approach has many advantages with regard to its generality, flexibility and interpretability that we believe will provide a great backdrop for stimulating discussions at NeurIPS 2020.

**Minor points.** We thank the reviewers for their suggestions of **prior work**, which we now cite and discuss. R2 asked **how a non-differentiable system (old Fig. 4) could be trained**. - We previously used finite differences to compute gradients of the loss w.r.t. plasticity parameters, using perturbations that change the number of output spikes in at least one data set. We now also use CMA-ES where a selection mechanism governs plasticity parameters updates instead of gradient descent. R4 commented on **the plausibility of the spiking set-up** (old Fig. 4). - While it is not realistic to assume inhibitory plasticity acts in isolation, we chose a simple rule as a first goal in a spiking network. It is a rule ascribing a given function to a network, stable and analytically tractable. R4 will hopefully be relieved to know that including excitatory plasticity is currently in progress and we will report our progress for combined plastic inhibitory and excitatory synapses in the camera ready version.

Figure 1: **Extended results** for old Fig. 1, with $N = 100$ inputs (A) and Fig. 3, with $N = 50$ inputs and $M = 5$ outputs (B), using CMA-ES instead of SGD + finite differences. The same color scheme and notations as in the original figures were used. In (B) the parameters of the two co-optimized rules have been collapsed in a single plot.

[Meta-Review · NeurIPS 2020]

Despite the reviewers are still concerned with respect to some issues, like the power of discovering new plasticity rules or the lack of empirical evidence on real data, the rebuttal provided a satisfactorily answer to the criticism of scalability.